# Parameter estimation for ocean background vertical diffusivity coefficients in the Community Earth System Model (v1.2.1) and its impact on ENSO forecast

Zheqi Shen[1,2,4], Yihao Chen[2], Xiaojing Li[3,4], and Xunshu Song[3,4]

[1]Key Laboratory of Marine Hazards Forecasting, Ministry of Natural Resources, Hohai University, Nanjing, China
[2]College of Oceanography, Hohai University, Nanjing, China
[3]State Key Laboratory of Satellite Ocean Environment Dynamics, Second Institute of Oceanography, Ministry of Natural Resources, Hangzhou, China
[4]Southern Laboratory of Ocean Science and Engineering (Zhuhai), Zhuhai, China

**Correspondence:** Zheqi Shen (zqshen@hhu.edu.cn)

**Abstract.** This study investigates parameter estimation (PE) to enhance climate forecasts of a coupled general circulation model by adjusting the background vertical diffusivity coefficients in its ocean component. These parameters were initially identified through sensitivity experiments and subsequently estimated by assimilating the sea surface temperature and temperature-salinity profiles. This study expands the coupled data assimilation system of the Community Earth System Model (CESM) and the ensemble adjustment Kalman filter (EAKF) to enable parameter estimation. PE experiments were performed to establish balanced initial states and adjusted parameters for forecasting the El Niño-Southern Oscillation (ENSO). Comparing the model states between the PE experiment and a state estimation (SE) experiment revealed that PE can significantly reduce the uncertainty of these parameters and improve the quality of analysis. The forecasts obtained from PE and SE experiments further validate that PE has the potential to improve the forecast skill of ENSO.

## 1  Introduction

The Coupled General Circulation Model (CGCM) is a prominent tool widely utilized for predicting future climate. However, limitations arise in simulations and forecasts derived from CGCMs due to imperfections in model physics, numerical schemes, and initial conditions. Coupled data assimilation, referred to as state estimation (SE), can enhance the accuracy and consistency of initial conditions in CGCMs by integrating coupled models with available observation data. SE is commonly employed in current operational forecasting systems (Stammer et al., 2016; Balmaseda et al., 2009). In addition to SE, researchers have developed parameter estimation (PE) or parameter optimization (PO) methods to mitigate model errors arising from uncertainties in empirical parameters of diverse physical parameterization schemes. These methods optimize model parameters using observation data, leading to a substantial reduction in model errors (Evensen et al., 1998; Zhang et al., 2020).

Recent studies have shown the potential of PE to enhance forecast accuracy (Wu et al., 2012; Zhang et al., 2012) by reducing model biases (Tong and Xue, 2008a, b). For instance, PE experiments have been performed using conceptual models (Han et al., 2013), intermediate complexity models (ICM) (Wu et al., 2016), and CGCMs (Liu et al., 2014a) to illustrate the capacity of

PE to address model errors and enhance the predictability of climate and weather events. However, most of these studies were carried out under perfect model scenarios, and only a limited number of studies have estimated parameters using real observation data. PE encounters various challenges in real-world scenarios, including inconsistencies in initial conditions, biases in numerical models compared to reality, and difficulties in determining the ideal value of unknown parameters (Zhao et al., 2019).

Despite these challenges, several examples of PE in actual forecast models exist. For example, Menemenlis et al. (2005) used Green's function approach to estimate parameters in an ocean general circulation model, demonstrating improved estimations compared to the prior values. Hu et al. (2010) performed parameter estimation in a weather model, confirming that optimized parameters can improve the model's forecast accuracy for real-world weather events. Kondrashov et al. (2008) used observation data to estimate parameters in a simplified ICM, verifying that optimized parameters can better match observation results. Similarly, Zhao et al. (2019) and Gao et al. (2021) performed parameter estimation in the Zebiak-Cane model, another ICM, using real observations. And they both revealed that prediction skills for ENSO were improved with the estimated parameters.

There is still challenges utilizing PE with observation data in CGCMs for the purpose of improving forecasts and reanalysis (Zhang et al., 2020). To overcome these challenges, we employed the improved PE method proposed by Shen and Tang (2022) in this study to estimate background vertical diffusivity coefficients in the ocean model via PE experiments. A coupled data assimilation system that is built upon the Community Earth System Model (CESM) and the ensemble adjustment Kalman filter (EAKF) method (Anderson, 2003) is used in this study. Specifically, we assimilated various ocean observation data, such as satellite sea surface temperature (SST) and temperature-salinity (T-S) profile data, to provide optimal parameters for seasonal forecasting. Additionally, we used the results obtained from PE to initialize ENSO forecasting, which were subsequently compared to SE initialization results to illustrate the advantages of PE in enhancing the CGCM's ENSO forecasting ability.

The paper is organized as follows: Section 2 introduces the data assimilation system, observation data, PE method, and experimental settings. Section 3 presents the results of the sensitivity experiment while comparing the analyses and forecasts using PE and SE. Lastly, Section 4 concludes the study.

## 2  Data assimilation system and PE methods

It has been demonstrated that coupled models can provide more compatible initial conditions via coupled data assimilation (Fujii et al., 2009; Mulholland et al., 2015; Penny and Hamill, 2017), which, in turn, improves seasonal predictions (Jin et al., 2008; Kug et al., 2008). In a previous study, we developed a coupled assimilation and ensemble forecasting system based on the fully coupled model CESM (Chen et al., 2022). The system employed the EAKF method to assimilate ocean observations from various sources and adjust the state variables of the ocean model, thereby influencing other model components through flux exchanges in the coupled process. Notably, the assimilation results has been demonstrated significant improvement for ENSO forecast skill (Chen et al., 2023). In the current study, we extended this system by incorporating a parameter estimation function, which enabled the estimation of several critical parameters in the ocean model.

## 2.1 The CESM model and the background vertical diffusivity coefficients

The study utilized version 1.2.1 of the open-source global coupled model, CESM, developed by the National Center for Atmospheric Research (NCAR). This integrated model includes the Community Atmospheric Model version 4 (CAM4) (Neale et al., 2010), the Parallel Ocean Program version 2 (POP2) (Danabasoglu et al., 2012), the Community Ice Code version 4 (CICE4), the Community Land Model version 4 (CLM4), as well as other modules. The atmospheric component has a horizontal resolution of $0.9° \times 1.25°$ with 26 vertical levels, while the ocean component was integrated at a nominal resolution of $1°$ with an enhanced meridional resolution of $0.5°$ in the equatorial region and 60 vertical levels.

In many OGCMs, vertical mixing can be parameterized separately by region, including upper boundary layer schemes and a diapycnal mixing scheme for the ocean interior. The K-profile parameterization (Large et al., 1994) is widely used to parameterize vertical mixing in ocean models. It includes a background diffusivity parameter that determines the diapycnal mixing in the thermocline. It is critical to the heat transfer between the upper boundary layer and the ocean interior. The background diffusivity parameter is typically set to a constant value, and its magnitude is determined by fitting the model to observations or theoretical considerations. As identified by much of the previous work, the background diffusivity parameterization is a key factor in vertical mixing parameterizations, and it has significant uncertainties and contributes to a large bias in SST simulations (Jochum, 2009). Zhu and Zhang (2018) have shown that a better background diffusivity parameterization leads to more realistic simulations of the cold tongue and equatorial thermocline, which has the potential to affect the fidelity of simulated seasonal to interannual variability in the tropical Pacific, such as the ENSO phenomenon. Therefore, the present study focused on estimating the parameters in background diffusivity parameterization.

The ocean model of CESM, POP2, was initially proposed by Smith et al. (1992) to solve three-dimensional ocean dynamic primitive equations on a global grid under the assumptions of Boussinesq and hydrostatic approximation. The background diffusivity parameter, denoted as $k_w$, is mainly used to characterize mixing processes resulting from the breaking of inertial internal waves (Smith et al., 2010). However, due to the uncertain propagation and dissipation behavior of these waves, the parameter value of $k_w$ has significant uncertainty. Munk (1966) first estimated the averaged diapycnal diffusivity of $10^{-4} m^2/s$ based on the advective-diffusive balance, thus requiring a background diffusivity of $O(10^{-4} m^2/s)$ to realistically produce the pycnocline in numerical models (Bryan, 1987). However, microstructure measurements generally give estimates that can be an order of magnitude reduced (Gregg, 1977; Ledwell et al., 1998). Hence, a constant background diffusivity of $O(10^{-5} m^2/s)$ has been typically applied in many ocean and climate modeling. Recently, observational evidence indicates that the assumed constant background diffusivity is not uniform but is spatially varying (Huussen et al., 2012; Kunze et al., 2006). In particular, microstructure measurements suggest that background diffusivity should be reduced near the equator, with a magnitude of $O(10^{-6} m^2/s)$ (Cheng and Kitade, 2014; Gregg et al., 2003). Jochum (2009) adopted a latitudinal structure of $k_w$ in the POP2 model and found that it simulated the ocean state better, which has been subsequently accepted by follow-up studies. Specifically, POP2 utilizes four independent background vertical diffusivity coefficients (BVDCs) to simulate the latitudinal structure of the background diffusivity. Table 1 lists each coefficient's default values and descriptions (Smith et al., 2010).

**Table 1.** Background vertical diffusivity coefficients (BVDCs) of KPP parameterization in POP2.

| Name list variable | parameter description | default value($cm^2/s$) | notations |
|---|---|---|---|
| bckgrnd_vdc1 | Background diffusivity | 0.16 | $v_1$ |
| bckgrnd_vdc_eq | Equatorial diffusivity | 0.01 | $v_e$ |
| bckgrnd_vdc_psim | Maximum PSI-induced diffusivity | 0.13 | $v_p$ |
| bckgrnd_vdc_ban | Banda Sea diffusivity | 1.0 | $v_b$ |

The BVDCs have a total of four constants, comprising a coefficient $v_1$ describing global diffusivity, a coefficient $v_e$ describing equatorial diffusivity (Gregg et al., 2003), a coefficient $v_p$ describing the diffusivity near 28.9°S and 28.9°N latitudes, and a coefficient $v_b$ describing diffusivity in the Banda Sea region alone(Jochum and Potemra, 2008). Among them, $v_p$ is also known

as the maximum PSI-induced diffusivity, representing the result of the parametric subharmonic instability (PSI) of the M2 tide (MacKinnon and Winters, 2005). Therefore, in POP2, the background diffusivity parameter $k_w$ has only a fixed form of spatial variation (Smith et al., 2010). Except for the Banda Sea, which takes on a specific value, $k_w$ globally varies only latitudinally and can be expressed as follows:

$$k_w = v_e + v_1(\frac{\theta}{10})^2 + v_p e^{-0.4(\theta+28.9)} + v_p e^{-0.4(\theta-28.9)} \tag{1}$$

where $\theta$ is the latitude.

Therefore, the $k_w$ value has been modified from its typically constant value of $0.1cm^2/s$ to $0.17cm^2/s$ ($v_1 + v_e = 0.17$) almost everywhere in POP2. And there are regions where different values are used: $1.0cm^2/s$ in the Banda Sea ($v_b$), $0.3cm^2/s$ in the latitude bands around $28.9°N/S$ ($v_1 + v_e + v_p$), and $0.01cm^2/s$ at the equator ($v_e$).

## 2.2   The data assimilation and ensemble prediction system

The Data Assimilation Research Testbed (DART) was employed in this study to implement the data assimilation system (Anderson et al., 2009; Karspeck et al., 2018). DART, an open-source software, offers various filter methods' implementations. Previously, this data assimilation system was used to study the impact of initial state errors on assimilation quality by assimilating ocean observations within a quasi-weakly coupled data assimilation framework (Chen et al., 2022). The EAKF method yielded the analysis ensemble, serving as the initial condition for climate variability forecasting. Notably, the system's ini-

tial conditions facilitated the forecasting of significant climate variability such as ENSO and IOD, and consequently directed toward a demonstrable improvement of forecasting skill (Chen et al., 2023).

The description of the assimilation system is presented in detail by Chen et al. (2022). In brief, it is a weakly coupled data assimilation system since only ocean observations are assimilated and the coupled model in used for integration. This study utilizes the ensemble adjustment Kalman filter approach with 20 ensemble members. The ensemble members are constructed

using long-term spin-up integration results and then repeatedly assimilating the WOA18 (Garcia et al., 2019) climatology data over 4 years to correct the climatological bias. This approach is essential to ensure the initial ensemble can effectively incorporate all observations during the data assimilation procedure.

Two sets of observation data are assimilated every ten days. One dataset is the optimum interpolation sea surface temperature (OISST) dataset version 2.1 retrieved from the National Oceanic and Atmospheric Administration (NOAA). The other is the EN4 profile dataset version 4.2.1 of the UK Met Office. The OISST dataset has a daily $0.25°$ resolution and was constructed by combining observations from different platforms (satellites, ships, buoys, and Argo floats) on a regular global grid. The EN4 profile dataset is a collection of ocean temperature and salinity (T-S) profiles obtained across global oceans from 1900 to the present. Quality control methods ensure good quality (Gouretski and Reseghetti, 2010).

The datasets were pre-processed before being assimilated into the system. Regarding the data assimilation system that assimilates SST and T-S profiles every ten days, daily profiles were merged and assigned to the final day of each sequence. To prevent overfitting due to assimilating excessive profile observations, the data at different depths were first interpolated to 31 layers from 5 m to approximately 2100 m and then averaged horizontally. Specific vertical depths were obtained from the EN4 analysis data (Good et al., 2013). The mean value of all data in each $1° \times 1°$ cell at each level was regarded as the observation value. Moreover, the OISST data were thinned such that only data on the $1° \times 1°$ grids were assimilated every ten days. Previous studies have shown that this processing method can produce effective state estimation results (Chen et al., 2022, 2023).

Localization was employed using the Gaspari and Cohn function (Gaspari and Cohn, 1999), which employed a cutoff half-width of 0.1 rad (approximately 600 km) for both observations. The SST and T-S profiles had vertical localization half-widths of 250 m and 1000 m, respectively. Additionally, the application of covariance inflation involved utilizing a constant inflation factor with $\alpha = 1.02$ for model states. These factors were determined empirically and verified in prior studies (Shen and Tang, 2022; Chen et al., 2022, 2023).

## 2.3 Parameter estimation method

One approach to achieving PE is the state vector augmentation method, in which parameters are treated as specific model variables and included in the state vectors. By updating the augmented state vector with observations, the model state and parameters can be estimated concurrently (Kivman, 2003; Annan and Hargreaves, 2004; Annan, 2005). Applying PE in a CGCM encounters several technical challenges. Firstly, many parameters in the GCM emanate from simplifying underlying physical processes, which may display globally uniform values. Updating a few global parameters with numerous data may accumulate sample errors, leading to PE failure. Since the parameters estimated in this paper are four constants, this is the main challenge in the experiments. To overcome this hurdle, we used the adaptive spatial averaging (ASA) method designed by Liu et al. (2014b) for CGCM. In each data assimilation step, we transformed each parameter from a single scalar value into a two-dimensional field, considering spatial dependence and localization during the assimilation. Afterward, we use the adaptive algorithm to average the two-dimensional parameter fields, to produce a scalar value incorporated in subsequent model integration. This algorithm calculates the ratio of the a posteriori standard deviation to the a priori standard deviation at each

grid point after each update of the two-dimensional parameters, which implies the strength of the effect of assimilation, and then averages the parameter values at grid points where the ratio exceeds a certain threshold. This threshold is chosen using an adaptive algorithm to ensure that a certain number of grid points (in this experiment 10,000 out of a total of 80,000 grids) are included in the calculation of the averaged parameters. More details refers to Shen and Tang (2022).

A further challenge arises from covariance inflation. Studies have noted that the parameter ensemble's spread (standard deviation) is generally relatively lower than that of the state ensemble, primarily because parameters remain constant for the mode integration. Consequently, ensemble Kalman filter-based PE requires a larger covariance inflation factor for the parameter ensemble. In a previous study, we employed twin experiments to demonstrate the necessity of covariance inflation for PE of BVDCs in CESM and developed a two-stage covariance inflation approach (Shen and Tang, 2022). Specifically, the conventional covariance inflation was applied to the augmented vector of the model states and the 2-D parameter fields, using a fixed inflation factor of $\alpha = 1.02$ before assimilation. Afterward, we average the analysis data of 2-D parameter fields to obtain global scalars and utilize a covariance inflation factor of $\alpha_p = 1.25$ solely for the parameter ensemble. This factor is deduced by calculating the average growth rate of the state variables in the model integration. Figure 1 provides a schematic diagram illustrating the PE process described above.

However, the conditional covariance inflation (CCI) method is usually used in practical data assimilation to ensure that the ensemble spread does not fall below a lower bound (Aksoy et al., 2006; Liu et al., 2014a). The CCI is designed to inflate the parameter ensemble spread back to a predefined threshold value when it is smaller than the threshold. In this work, we intentionally do not use the CCI method and let the parameter ensemble degenerate after several data assimilation cycles. After that, all ensemble members use the same improved parameters that no longer changes value with the subsequent data assimilation. This strategy allows the parameter ensemble to converge and makes subsequent ensemble forecasting experiments easier to implement.

It is also worth noting that in order to avoid unphysical parameter values, after each parameter estimation, if an abnormal parameter value (e.g., negative value) occurs for an ensemble member, we remove the parameter and use the parameter of the neighbouring member to integrate the model.

## 2.4 Experimental design and verification data

Conducting sensitivity analyses (Navon, 1998) before PE is necessary to ensure the parameters have significant impact on the observed variables. In this study, the sensitivity experiment was initially conducted to show the sensitivity of model temperature and salinity to the BVDCs. An ensemble of size 20 was integrated using the same initial states but with perturbed parameters. We perturbed each BVDC by adding noise generated from a Gaussian distribution with a mean value of 0 and a standard deviation of 30 percent of its default value. We measured the variable sensitivity to the perturbed parameters by using the ensemble spread of each variable.

Subsequently, we conducted separate SE and PE experiments using the initial ensemble introduced earlier. The assimilation time window started in January 2005 and continued until December 2017. In the SE experiment, the SST and T-S profiles were assimilated every 10 days to update the model state variables that include temperature, salinity, sea surface pressure, and surface

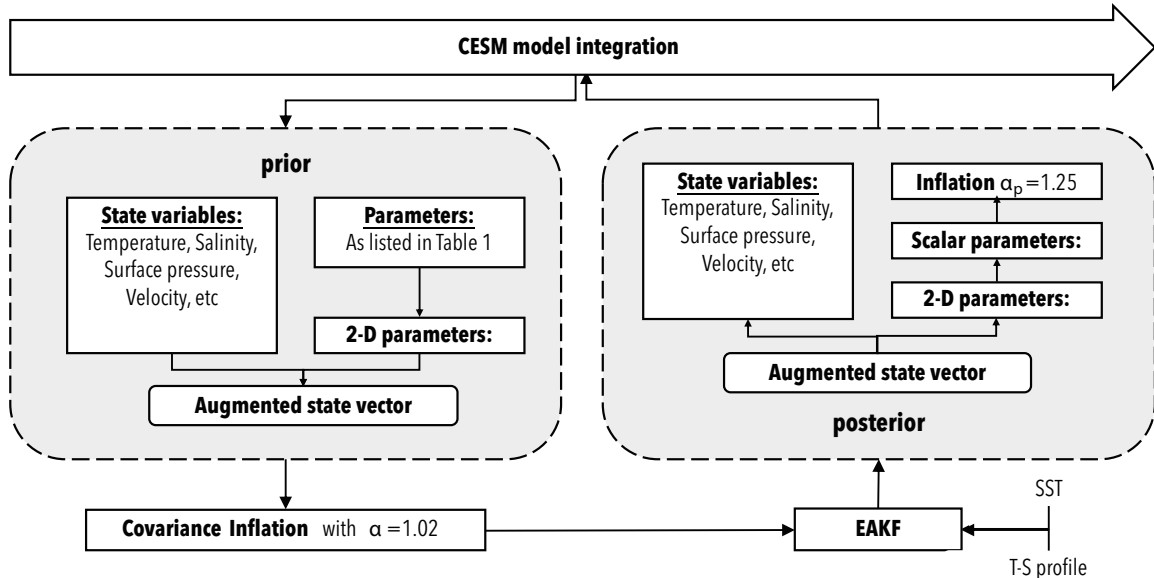

**Figure 1.** Schematic diagram of the parameter estimation process in the CESM model.

current velocity. The BVDCs listed in Table 1 were used in model integration during the entire period. The PE experiment used the same observations to update the model state variables and the BVDCs concurrently. As Zhang et al. (2012) showed, the

signal-to-noise ratio of the state-parameter error covariance in the coupled model can be significantly improved after the state estimation reaches quasi-equilibrium. Therefore, we performed only pure state estimation in the first year of the PE experiment and activate the PE function from the beginning of the second year. That is, the parameter values change gradually from 2006 onwards. At this point, the observation-constrained states can improve the parameter estimates more effectively.

    We compared the results of SE and PE experiments with validation data to demonstrate the impact of PE on reducing analysis

errors. The temperature and salinity from the objective analysis data of EN4.2.1 (Good et al., 2013) are used for validation. It should be noted that the EN4 profile dataset for assimilation is a collection of profiles, and the EN4 objective analysis dataset is processed and gridded data. To ensure impartiality in the validation data, we also incorporated high-quality reanalysis products such as ORAS4 by Balmaseda et al. (2013) and GFDL/ECDA by Zhang et al. (2007).

    The EAKF can provide initial conditions for ensemble prediction by running an ensemble of members. The analysis ensem-

bles of SE and PE experiments were utilized as initial conditions for climate forecast with the coupled model. We conducted ensemble forecast experiments from 2008 to 2017, using the analysis ensembles derived from both SE and PE. The parameters obtained by PE were also employed in the latter case. Predictions were issued at the beginning of each January, April, July, and October, extending for 12 months. The Hadley Centre Sea Ice and Sea Surface Temperature dataset (HadISST) (Rayner et al., 2003) served as a reference dataset to compare the produced prediction products.

The schematics in Figures 2a-c show the sensitivity experiment, the SE experiment, and the PE experiment, respectively. It can be seen that the sensitivity experiment is a free integration experiment using the same initial condition and different

parameters. The SE experiment uses the ensemble of state variables and the same default parameters. At the same time, the PE experiment uses ensembles for both state variables and parameters. PE experiments are divided into three phases, which we will specify in the Results and Discussions section. Moreover, it also shows that the state and parameter estimation results are used in the later hindcast experiments.

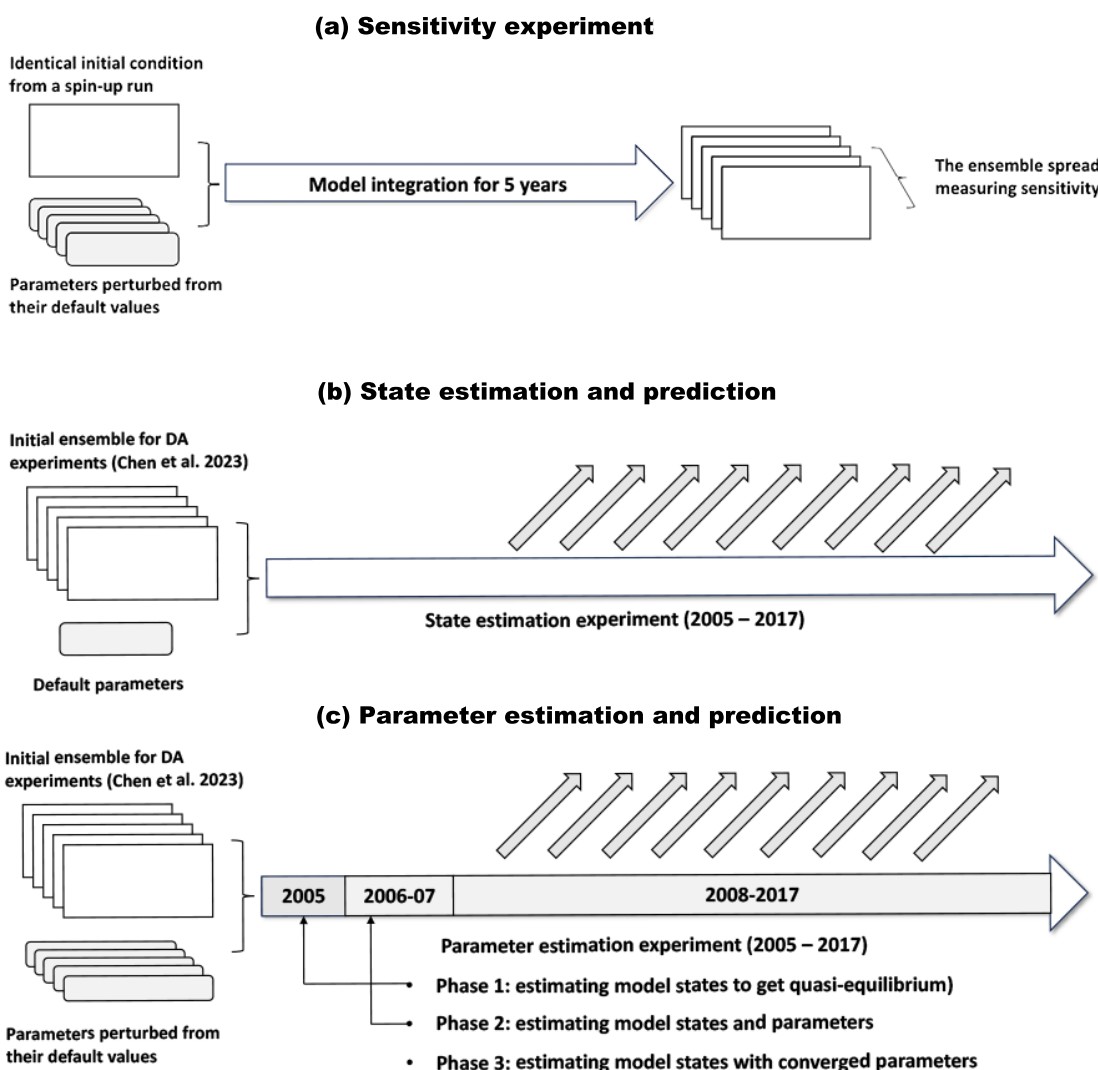

**Figure 2.** The schematic diagrams of (a) the sensitivity experiment , (b) state estimation and prediction experiment, (c) parameter estimation and prediction experiment.

## 3 Results and discussions

### 3.1 Sensitivity experiment

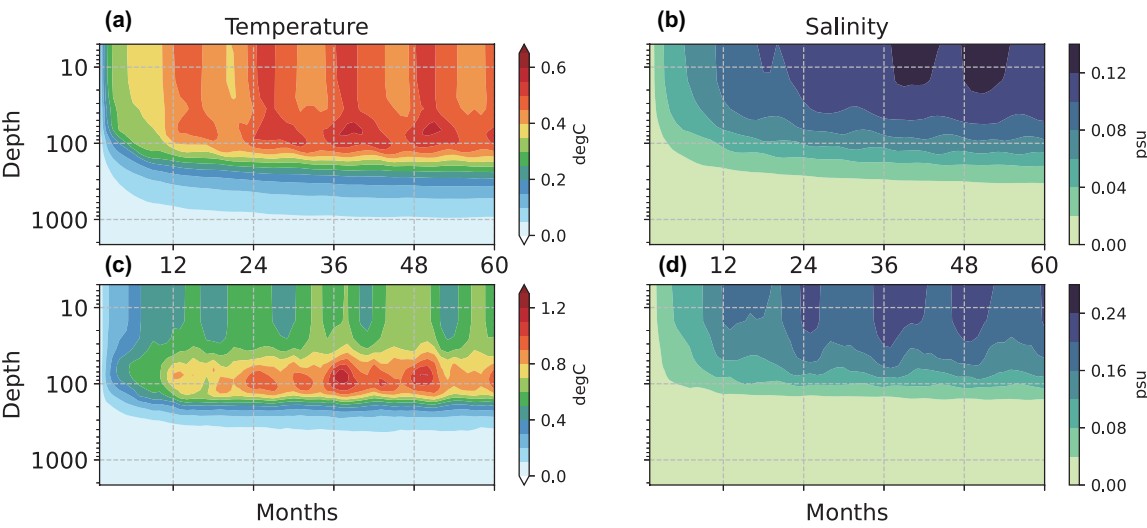

**Figure 3.** The global averaged (between $60°$S and $60°$N) ensemble spread of temperature (a) and salinity (b); (c) and (d) are the same as (a) and (b), but were averaged over the equator (between $5°$S and $5°$N)

Twenty identical ensemble members were utilized for the parameter sensitivity experiment in the CESM integration, which persisted for five years, with four parameters perturbed simultaneously. It is worth noting that we tried experiments where we perturbed the parameters one by one, and the experimental results showed that perturbing the different BVDCs had a comparable effect (not shown here), so we used this simultaneous perturbation scheme. The ensemble spreads of temperature and salinity variables, which measures their sensitivity to the perturbed parameters, are shown in Figure 2. The global ($66.5°$S - $66.5°$N) and equatorial ($5°$S - $5°$N) averaged temperature and salinity ensemble spreads were demonstrated accordingly. Perturbing BVDCs in the model leads to a rapid increase in temperature and salinity ensemble spread within the first year, followed by relative stability in the succeeding years. Figure 3 displays that temperature variables have the maximum sensitivity to BVDCs at approximately 100 meters depth, with salinity variables being most sensitive to these parameters at the sea surface. The influence of parameter's uncertainty can extend up to a depth of approximately 400 meters. Additionally, the equatorial area is highly sensitive to BVDC parameters in temperature at a depth of 50-100 meters and in salinity at the sea surface. The surface temperature variability in Figure 3a shows a conspicuous seasonal cycle which can possibly be related to the diverse rates of change in mean temperature instigated by distinct ocean areas between the northern and southern hemispheres.

The last three years' outcomes were used to computed the mean spread and analyze its spatial distribution. Figure 4 provides additional validation that temperature variability is highest within the equatorial range and most pronounced at a depth of 100m. In deeper layers, the parameters affect the temperature more significantly in western Pacific. Additionally, salinity is

highly sensitive to the parameters in the warm pool region of the tropical western Pacific, and the sensitivity of salinity to parameters is highest in the shallow layers, less than 50m depth. Furthermore, in extratropics, the temperature and salinity in the Kuroshio extention and Gulf Stream regions are also sensitive to these parameters to some extent.

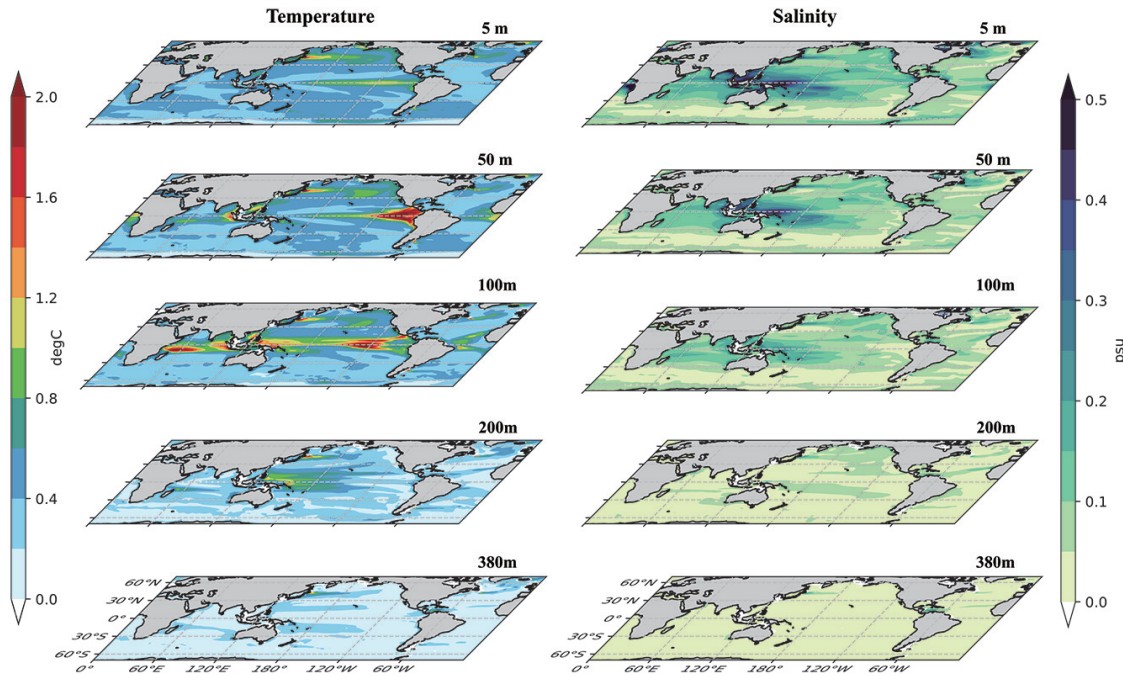

**Figure 4.** Spatial distribution of the ensemble spread of sea temperature (left) and salinity (right) at different depths.

The sensitivity experiment shows that the model temperature and salinity are sensitive to the uncertainty in the BVDCs, strongly indicating that assimilating SST and T-S profiles can potentially reducing the uncertainty.

### 3.2 Estimated parameters

We conducted separate SE and PE experiments, assimilating observations during the period between January 2005 and December 2017. In the SE experiment, default values of the BVDCs were consistently used in all ensemble members throughout the entire period. However, the PE experiment comprised three distinct phases. During the initial phase, we utilized perturbed parameters to perform state estimation. It spans a period of 1 year and brings the state estimation process to approximately quasi-equilibrium, where the uncertainty of coupled model states is sufficiently constrained by observations. In the second phase, spanning from 2006 to 2007, we activated the PE function illustrated in Figure 1. This function facilitated continuous correction of the parameter ensemble through observations. Finally, during the third phase, spanning from 2008 to 2017, these parameters remained unchanged.

Figure 5 depicts a graphical representation of the 20 ensemble members of the four BVDCs over time, with the ensemble mean represented by the red solid line. Observations gradually decreased the spread of the parameter ensemble, resulting in less uncertainty. After approximately two years, the parameter ensemble degenerated, and the spread reached 0. Consequently, assimilating observations could no longer adjust the parameters.

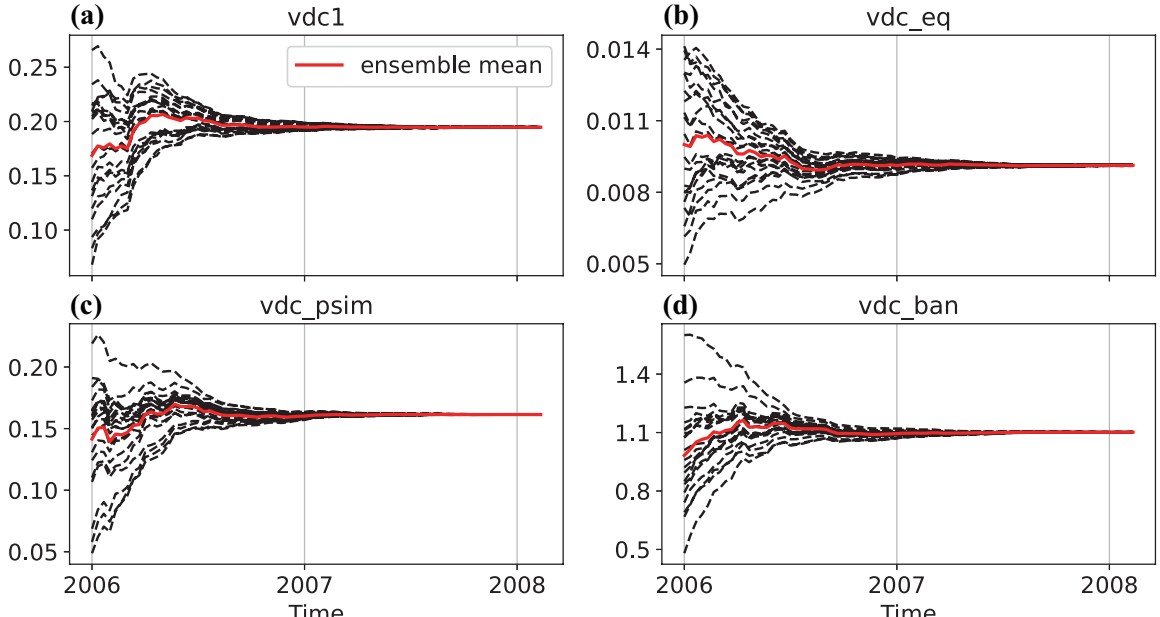

**Figure 5.** Evolution of each parameter since 2006, in which the red solid lines indicate the ensemble mean.

**Table 2.** PE final BVDC values.

| Parameters | $v_1$ | $v_e$ | $v_p$ | $v_b$ |
|---|---|---|---|---|
| Default value($cm^2/s$) | 0.16 | 0.01 | 0.13 | 1.0 |
| PE final value($cm^2/s$) | 0.195 | 0.0091 | 0.161 | 1.10 |
| Ratio of increase | 21.9% | -9% | 23.8% | 10% |

Table 2 presents the final values of BVDCs. Notably, $v_1$ and $v_p$ values are 20% higher than the default values, while $v_b$ is 10% higher (except for $v_e$, which is slightly lower than the default value). It's also worth noting the almost globally increasing value of the background vertical diffusivity, $k_w$, as calculated through Eq. 1 and depicted in Figure 6. The left-hand side of Figure 6 displays the band structure of the default background diffusivity, while the increment obtained by PE is shown on the right-hand side, further validating the achieved results.

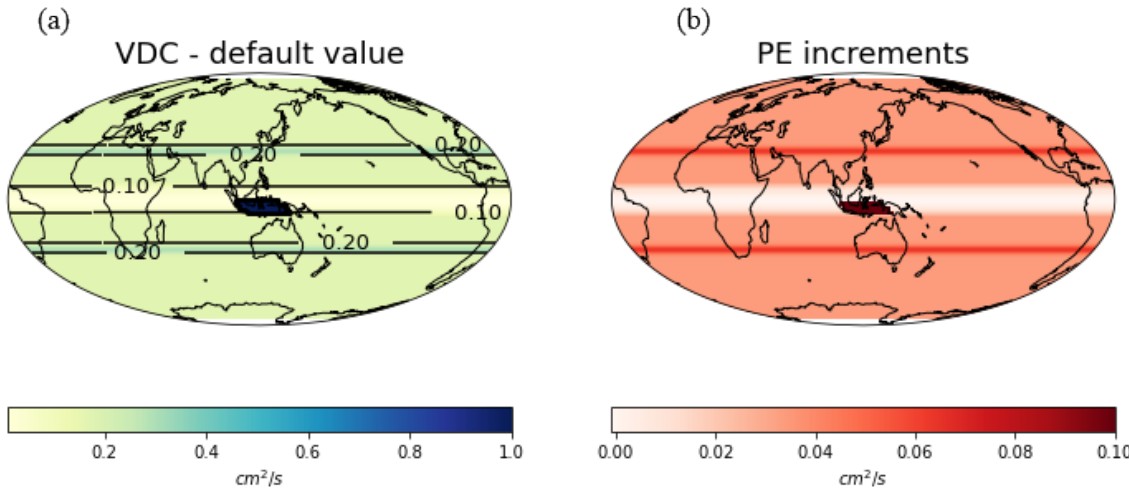

**Figure 6.** (a) Default latitudinal structure of background vertical diffusivity; (b) the increment of the background vertical diffusivity from PE.

## 3.3 Quality of the analysis

As previously mentioned, the parameters have remained unchanged since 2008. Consequently, the third phase of the PE experiment can be considered a distinct SE experiment using the estimated parameters listed in Table 2. This study focuses
specifically on evaluating the analyses obtained from the third phase by comparing the results of the PE and SE experiments.

we compare our analysis fields with the gridded objective analysis data from EN4 and other reanalysis products to demonstrate the validity of our results. Figure 7 displays the root mean squared error (RMSE) of the temperature in the analysis fields for the period of 2008-2017 by region. We compared the results with EN4, ORAS4, and ECDA. The regions are Global (within 66.5°N-S), Pacific, Indian Ocean, Atlantic, and intra-tropical (within 30°N-S). Similar findings can be observed globally and
250 in most regions using different datasets. When examining the global mean temperature, the depths with significant analysis errors are consistent with the parameter-sensitive depths, indicating that parameter uncertainty can impact the analysis accuracy. Moreover, the reduced RMSEs of the PE experiment indicates that PE improves the quality of the analysis. In particular, noticeable improvements are observed below a depth of 100 m. The most pronounced improvement is observed in the Atlantic Ocean and in tropical regions. Figure 8 illustrates the salinity errors in the analysis. The highest error is observed in the sea
surface layer, which is consistent with the most sensitive depth to parameters (Figure 3b and d). In contrast to temperature, PE primarily enhances salinity accuracy in deep Atlantic and tropical regions.

Figure 9 displays the RMSEs of the SE experiment and EN4 data in the tropics while emphasizing the disparity with the PE experiment. As Figure 9a shown, the most considerable temperature errors appear in all oceans around the depth of 100 meters, which matches the sensitivity analysis result for temperature depicted in Figure 9c. Figure 9e denotes an improvement in PE
for these errors, implying its usefulness throughout the tropics. It is noted there is improvement in areas where the temperature

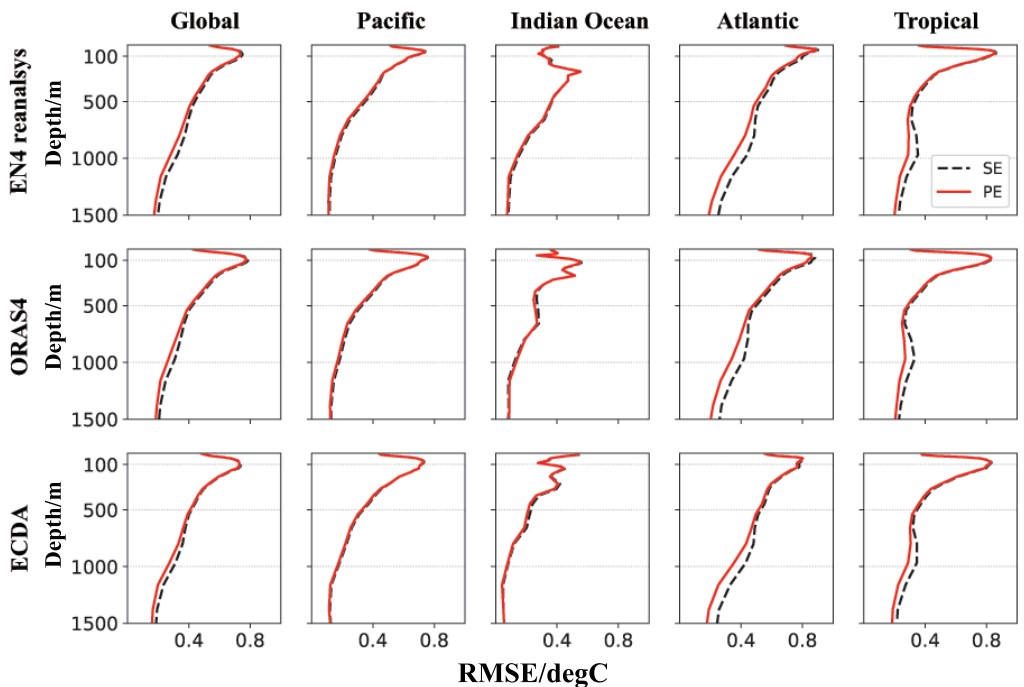

**Figure 7.** The temperature RMSE of the data assimilation results with EN4 (top), ORAS4 (middle), and ECDA (bottom) for the period of 2008-2017 by region. The regions are - Global (within 66.5°N-S), Pacific, Indian Ocean, Atlantic, and intra-tropical (within 30°N-S), respectively, from left to right.

are sensitive to the uncertainty of those parameters. And the temperature error in the deeper layers of the tropical oceans has also been reduced, especially in the deeper Atlantic Ocean. Figures 9b, 9d and 9f show the results for salinity. Although not as significant as temperature, the depths and areas where salinity errors emerge in the SE analysis align with those sensitive to parameters. Unsurprisingly, PE partially mitigates these errors, most significantly around the Andaman Sea, waters near Indonesia, and the coastal West African waters where vertical mixing is intensive.

The model bias of CESM is relatively large in the Atlantic Ocean. Danabasoglu et al. (2012) have shown the zonal-mean temperature and salinity of CCSM4 (which uses the same ocean model as CESM) minus climatology from observation. They noted that the deep Atlantic Ocean remains generally warmer than observed by about $0.58^oC$ in the mean. The local temperature and salinity maxima between $20°$ and $30°$N at a depth of about 1000 m are associated with the warmer and saltier than observed Mediterranean outflow through the Strait of Gibraltar. The largest salty biases occur in the deep Atlantic Ocean. The upper-ocean Atlantic north of $15^o$ N remains mostly saltier than the climatology. We show similar results in Figure 10, which displays the zonal-mean temperature and salinity of the Atlantic minus the climatology calculated from EN4 data using SE results and PE results, respectively. By comparing the climatology bias of the results from two experiments, it is seen that the most

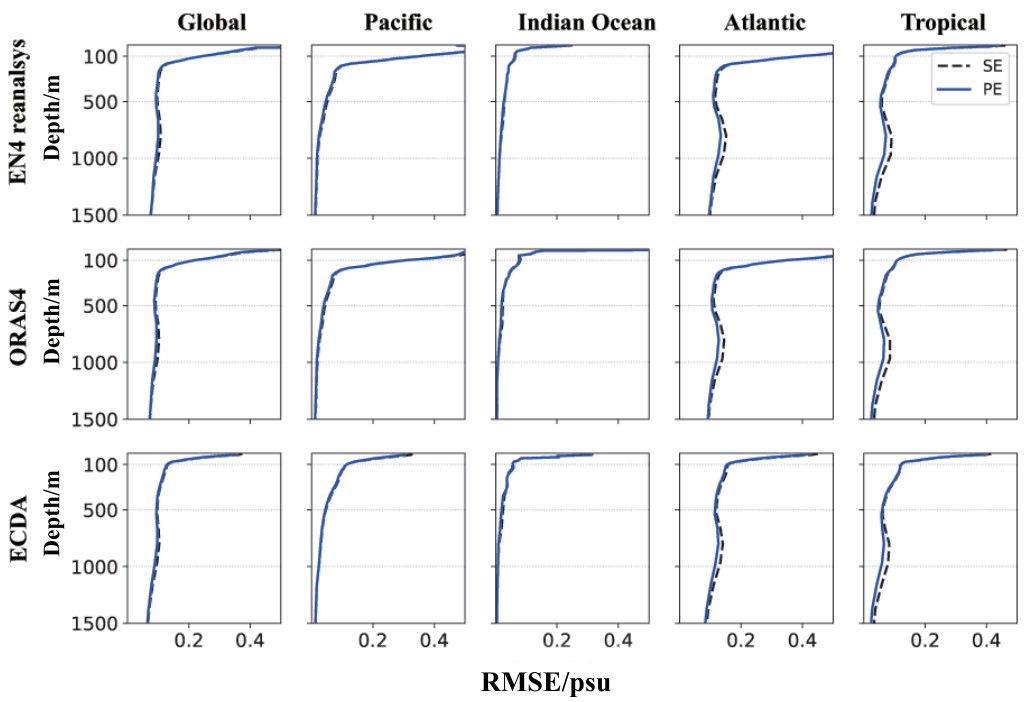

**Figure 8.** The same with Figure 6 but for salinity RMSE.

significant improvement of the PE on the SE lies in the Atlantic Ocean at a depth of 1000 m between 20° and 30° N latitude.
It strongly suggests the contribution of improved background diffusivity parameters to reducing model systematic biases. It
can be inferred from the conclusions of Danabasoglu et al. (2012) that this improvement also stems from the improvement of
the outward flow in the Strait of Gibraltar. Although the sensitivity and analytical errors in this region cannot be demonstrated
directly in Figure 9 due to resolution, the effect of PE is demonstrated by affecting the 1000m Atlantic Ocean between 20° and
30° N latitude. This explains the smaller Deep Atlantic RMSE in the PE results presented in Figures 7 and 8.

## 3.4 ENSO forecast experiment

This study utilized analysis ensembles from the coupled data assimilation system to conduct ENSO forecast experiments
between 2008 and 2017 (as shown in the schematic diagram in Figure 2b and 2c). The Nino-3.4 index, calculated as the
averaged sea surface temperature anomalies between the latitudes of 5°S to 5°N and longitudes of 190°E to 120°W was
employed to illustrate the variability of ENSO. The Nino-3.4 indices of the SE and PE forecasts were computed against various
lead times. The anomaly correlation coefficients (ACC) of these outcomes with the index derived from HadISST data were
employed to measure the prediction skills, as shown in Figure 11a. Moreover, Figure 11b depicts the RMSE of the forecasts
against HadISST.

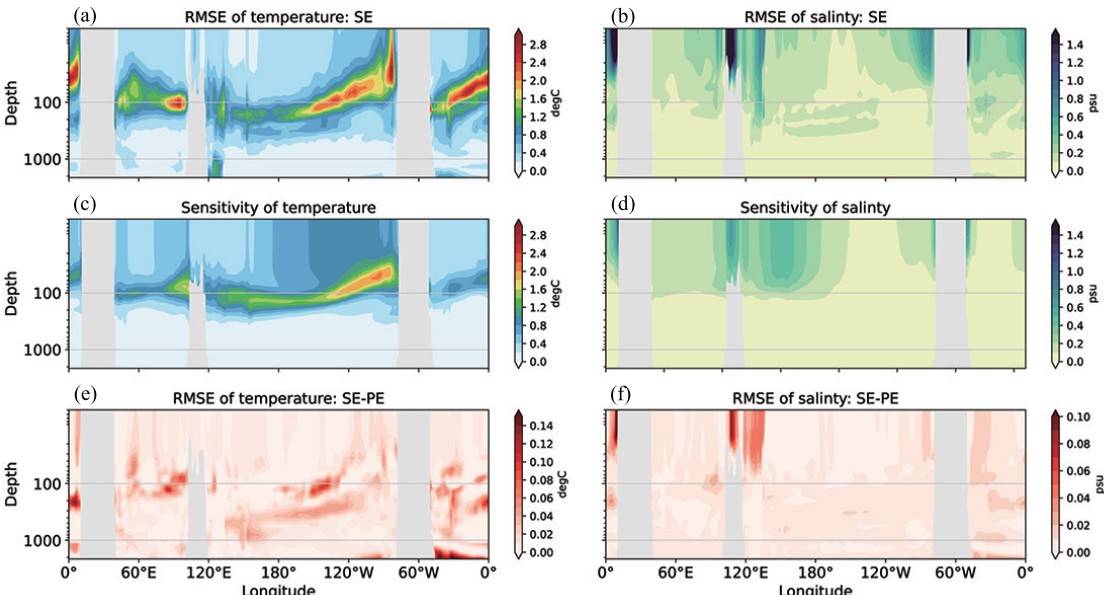

**Figure 9.** The temperature (a) and salinity (b) RMSE of the SE results and EN4 data in the tropics; the mean temperature (c) and salinity (d) spreads of the sensitivity experiment results in the tropics; the difference between the temperature (e) and salinity (f) RMSE of the SE results and that of the PE results, respectively.

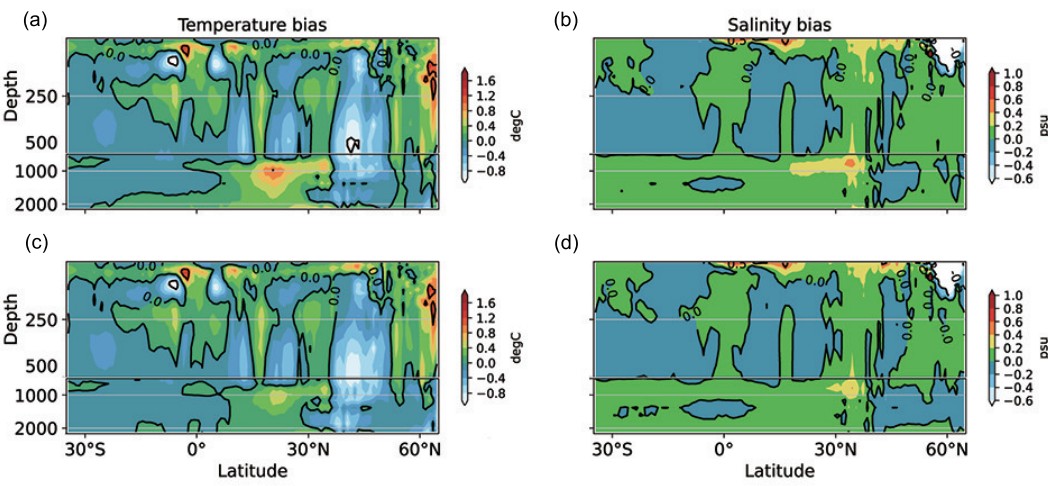

**Figure 10.** The zonal-mean temperature (a) and salinity (b) of the Atlantic minus the climatology calculated from EN4 data using the SE results, and the zonal-mean temperature (c) and salinity (d) of the Atlantic bias using the PE results.

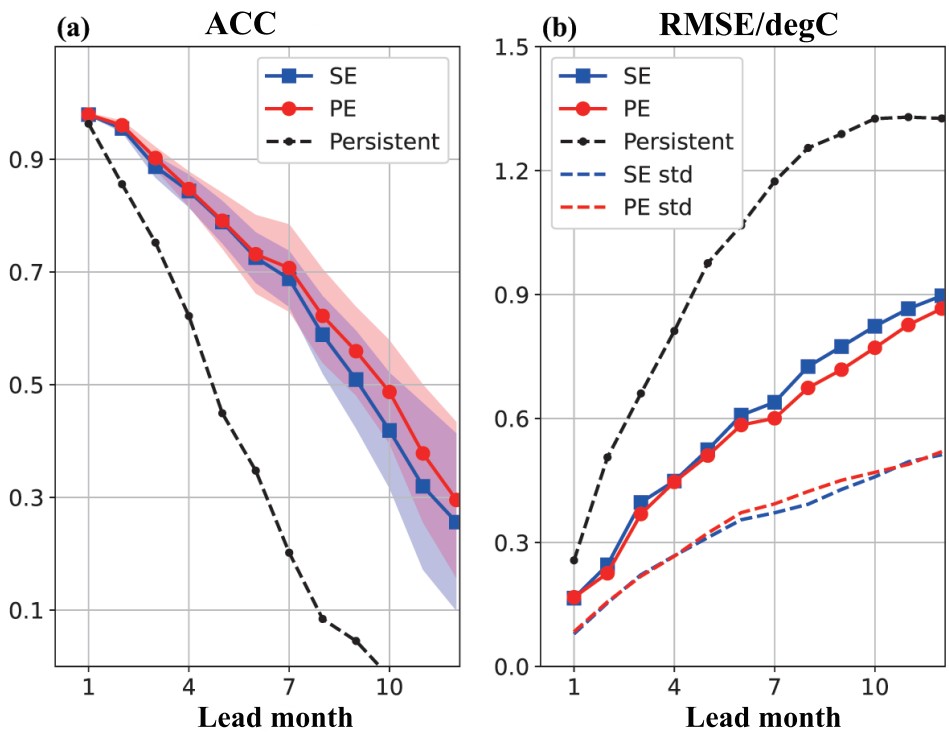

**Figure 11.** Correlation coefficients (a) and root-mean-square errors (RMSEs) and mean spreads (b) of the observed and forecasted Niño-3.4 index as a function of lead time.

The prediction skills of both SE and PE cases were significantly superior to those of the persistence skills represented by a black dotted line. For lead times exceeding five months, the PE case exhibited higher ACCs compared to the SE case. By setting the threshold value of an effective prediction as a ACC of 0.5, which is equivalent to the 99% statistical confidence level with an independent sample size of 30, it was observed that the SE case effective predict ENSO at a lead time of up to 9-month which is 1 month in advance compared to the PE case. To demonstrate the significance of the PE advantage, ACCs were computed for each ensemble member using HadISST. The shaded areas represent the ACCs of the ensemble mean plus/minus the standard deviation of the ACCs of each member, further confirming the superior prediction ability of the PE case.

The root mean square errors (RMSEs) of the PE case were also lower than those of the SE case, particularly after a lead time of 5 months. Additionally, the ensemble spreads (colorful dashed lines in Figure 11b) of the PE results were larger compared to those of the SE predictions. Since the spreads of the PE results were closer to the RMSE than those of the SE results, it indicates that the PE initial conditions are more consistent.

Figure 12 illustrates the spatial correlation coefficient pattern between the predicted sea surface temperature (SST) anomaly and the corresponding HadISST data over the tropical Pacific for the SE and PE cases. The SE and PE results showed no

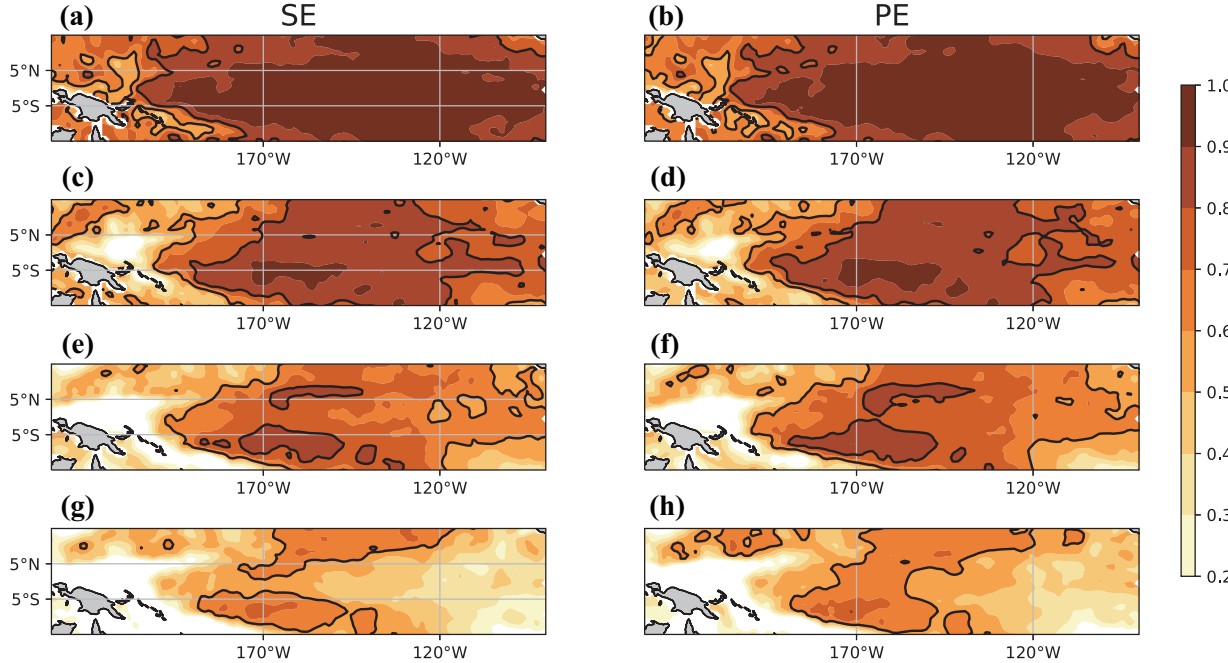

**Figure 12.** Spatial pattern of the correlation coefficients between the predicted and observed SST anomalies (SSTA) with SE initialization (left column) and PE initialization (right column) at the 1-month, 4-month, 7-month, and 10-month lead times.

significant difference for lead times of one or four months. However, for longer lead times, the initial conditions and parameters obtained through PE significantly improved the SST anomaly in the tropical Pacific. As illustrated in Figure 9, the outcomes of our data assimilation experiments reveal a notable reduction in analysis error within the thermocline of the equatorial Pacific due to parameter estimation. This yields improved initial conditions for ensemble prediction. Simultaneously, the refined parameters contribute to enhanced global vertical mixing. Consequently, the SSTA in the tropical Pacific, derived from our forecasts incorporating parameter estimation, exhibit a higher ACC with observational data. In addition, since the advantage of the PE analysis in the subsurface layer takes some time to affect the SST, the forecast skill based on the PE results is only significantly greater than that of the SE results at longer lead times.

The improvement in PE for ENSO forecasts is likely attributed to better simulations of subsurface temperatures. Previous studies have consistently shown that the accuracy of initial subsurface conditions is crucial for ENSO event prediction (Tang et al., 2003; Song et al., 2022). The PE method not only enhances the accuracy of the initial subsurface conditions (as demonstrated in Figure 7 and Figure 8), but also provides constrained parameters that more accurately represent the background diffusivity process in the ocean model, leading to improved forecast skills.

## 4    Conclusions

Errors in the coupled model can arise from uncertainties in the dynamic cores, numerical schemes, physical parameterization schemes, and empirical parameters. PE is the process of adjusting or optimizing model parameters using observations, the method of PE is very similar to SE. However, PE has additional complexity since parameters are indirectly related to model states, and the state-parameter covariance is challenging to estimate.

In this study, the fully coupled CESM was used to perform the SE and PE experiments, in which satellite SST and subsurface
T/S profiles were assimilated using an ensemble Kalman filter to estimate the model states and critical parameters in vertical mixing parameterization. The SE system was established and comprehensively evaluated by Chen et al. (2022), and PE methods were developed using a new solution to deal with constant parameter evolution (Shen and Tang, 2022). In this work, we used these systems to conduct experiments to compare the SE and PE in the CGCM.

The parameter sensitivity experiments were first conducted to evaluate the sensitivity of the model variables to the parame-
ters, which were measured by the ensemble spread for the temperature and salinity variables significantly. Figure 3 and figure 4 show that the BVDCs impact the model temperature and salinity variables significantly. Therefore, the PE is theoretically feasible using SST and T/S observations.

However, in this work, we intentionally do not use the CCI method and let the parameter ensemble degenerate in the PE experiment. At this point, all ensemble members can use the same improved parameters to carry out the ensemble forecast,
which makes the ensemble forecast easier to carry out and compare with other schemes. This may not be the optimal solution for parameter estimation, but it is the most convenient solution for carrying out realistic forecasts. The experiment in Shen and Tang (2022) also assimilates synthetic TS profiles and SST observations to estimate the vdc1 parameter in the same CESM model. As shown in their figure 10, the amount of change in the parameter values after a period of assimilation is very small as long as the proper parameter covariance inflation coefficients are used (scenarios a - c), so we believe that switching off the
CCI also yields relatively improved parameters.

The data assimilation results, using either SE or PE, were assessed against the EN4 objective analysis dataset and the other reanalysis datasets. The DA analyses errors (Figure 7 - Figure 10) and the parameter sensitivity results have similar patterns, ultimately revealing that the model errors were partly caused by uncertainties in these parameters. PE can reduce analysis errors in sensitive domains by considering the parameter uncertainties during assimilation.

One key challenge of using PE with real observations is the verification of the parameters, which cannot be observed. In this study, the estimated parameters and PE-derived initial conditions are employed to perform ensemble ENSO prediction. The prediction outcomes provide evidence of the benefits of using PE. Figures 11 and 12 present evidence that using more accurate initial conditions and better parameters through the PE method increases the prediction skill of ENSO, further verifies our conclusions.

This study brings forward the advancement of PE studies, from the perfect model Observing System Simulation Experiment (OSSE) scenario to real-world observations assimilation in CGCMs. The comparison between PE and SE highlights the potential of PE to improve coupled model reanalysis and prediction. However, the results in Figure 11 and Figure 12 indicate that PE

only slightly improves the prediction skill of this coupled prediction system. However, the prediction skill of ENSO is affected by many factors, such as predictability, in addition to the initial conditions and model errors (Liu et al., 2022). Therefore every

350 bit of improvement of the ENSO dynamical prediction skill is of some practical significance. Nevertheless, we will also pursue higher dynamical prediction skill in the future research.

In addition, to reduce the complexity of the problem, we only estimated four parameters in the vertical mixing parameterization in this study. However, many parameters in various physical processes exist that have impacts on the simulation and prediction of ENSO (Gao and Zhang, 2017; Zhao et al., 2019), which should be considered in future studies. Moreover, the

355 spatial distribution of the parameter sensitivity, as shown in figure 3, has not been used in the PE algorithm. This may serve as a potential strategy (Shen et al., 2022) to improve the efficiency of PE methods in CGCMs.

*Code and data availability.* The data used for assimilation and validation in this study can be accessed online from the following sources: World Ocean Atlas 2018 (WOA18) (https://www.ncei.noaa.gov/access/world-ocean-atlas-2018), Optimum Interpolation Sea Surface Temperature (OISST) (ftp://eclipse.ncdc.noaa.gov/pub/OI-daily-v2/NetCDF-uncompress), EN4 (https://www.metoffice.gov.uk/hadobs/en4/download-

360 en4-2-1.html), Hadley Centre Sea Ice and Sea Surface Temperature data set (HadISST) (https://www.metoffice.gov.uk/hadobs/hadisst), Geophysical Fluid Dynamics Laboratory's Ensemble Coupled Data Assimilation (GFDL/ECDA) (https://www.gfdl.noaa.gov/ocean-data-assimilation-model-output/), and Ocean Reanalysis System 4 (ORAS4) (https://www.ecmwf.int/en/research/climate-reanalysis/ocean-reanalysis).

The Community Earth System Model (CESM v1.2.1) and the Data Assimilation Research Testbed (DART), both utilized and modified in this study for parameter estimation, are archived on Zenodo under the DOI: 10.5281/zenodo.8115394. The repository also includes the

365 experiment results and the scripts for plotting.

*Author contributions.* ZS: Conceptualization, Methodology, Writing – original draft & editing. YC: Methodology, Experiment. XL: Writing – review & editing. XS: Methodology, Experiment.

*Competing interests.* The authors declare that they have no known competing financial interests or personal relationships that could have appeared to influence the work reported in this paper.

*Acknowledgements.* This study was supported by grants from the National Natural Science Foundation of China (42176003), the Fundamental Research Funds for the Central Universities (B210201022), the Jiangsu Provincial Innovation and Entrepreneurship Doctor Program (JSSCBS20210252). We thank the reviewers for their insightful comments and suggestions that greatly improved the manuscript.

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
