# Peer review of "Parameter estimation for ocean background vertical diffusivity coefficients in the Community Earth System Model (v1.2.1) and its impact on ENSO forecast"

_Geoscientific Model Development, 2023_

## Referee Comment (RC2)

**Review of "Parameter estimation for ocean background vertical diffusivity coefficients in the Community Earth System Model (v1.2.1) and its impact on ENSO forecast"**

This study investigates parameter estimation (PE) in improving climate forecasts of a coupled general circulation model by adjusting the background vertical diffusivity coefficients in its ocean component. Comparing the model states between the PE experiment and a state estimation (SE) experiment reveals that PE can significantly reduce the uncertainty of these parameters and improve the quality of analysis. The forecasts obtained from PE and SE experiments further validate that PE has the potential to improve the forecast skill of ENSO. The work is interesting and may has potential application in practice. However, there are some issues unclear in the manuscript. I would suggest that it should be published in **GMD**, subject to **major revisions**.

**1. Major concerns:**

(1) The manuscript focuses on improving the background vertical diffusivity and its impact on ENSO forecast. However, the background vertical diffusivity is only important in the deep ocean and the coefficients of the KPP dominate the mixing process of the upper ocean. Therefore, results show that the main differences between PE and SE are in the deep ocean, and the improvement on the upper ocean is not significant as the authors mentioned. From the text, the author pays more attention to the process of ENSO which is mainly associated with the upper ocean process. So why do the authors not adjust and optimize the parameters of KPP?

(2) The key issue of adjusting model parameters with real data is how to ensure the physical meaning of the parameters. The manuscript does not explain how to avoid the parameters from exceeding their physical values, specially for the spatial varying parameters.

(3) It is unclear how the adaptive spatial averaging is calculated. How big is the specific average bin? Does it cause spatially discontinuous?

(4) Table 1 involves 4 parameters and the use of parameters is regional. Do you only adjust the corresponding parameters in specific regions such as the equator and the Banda Sea?

(5) How are the 20 sensitivity experiments and other experiments set up and operated? A figure regarding this should be added. Based on Table 2, the parameter adjustment is quite small, and the parameters tend to reach a certain equilibrium state. However, the observations (number and locations) used each year are different. Why do the adjustment of parameters not change year by year?

(6) Line 215: "The highest error is observed in the sea surface layer, ...." I can't see the relevant results in Fig.7. The main differences are still at the depths of 600m-1200m in the Atlantic and tropical regions.

(7) Based on the sensitivity experiments, the main impact is in the upper 100m. However, from the assimilation results, the main differences are still at the depths over 600m in the tropical and Atlantic regions. Why is this?

(8) Line 221: "It is noted there is substantial improvement in areas where the errors are significant, except for the deep Atlantic Ocean". Estimated from the magnitude of SE-PE, the proportion of error reduction is less than 5%. It can't be said "substantial improvement".

(9) Fig.9 only mentions the differences in the Atlantic Ocean. How about the tropical ocean?

(10) The forecast of ENSO does improve to a certain extent, but the improvement is not significant (The change of RMSE is less than 0.1°). Shown From Fig.11, it seems that most differences locate in the equatorial region. It may connect to the reduction of diffusivity in the region (smaller mixing and higher SST?). The authors should give more explanations regarding this issue.

**2. Minor Concerns:**

(1) The assimilation just involves the ocean component. It seems not a couple assimilation system.

(2) In Fig.7, is it the ORAS5 or ORAS4?

(3)Line 116: "These factors were determined empirically and verified in prior studies." References should be added here.

(4) Units should be added in Figs.6, 7 and 10.

---

## Author Comment (AC1)

** reviewer's comments in black and italics Author's comments in red

Reviewer 1:

*This study presents parameter estimation experiments utilizing the CESM model to assimilate Sea Surface Temperature (SST) and Temperature/Salinity (T/S) profiles for initializing ENSO prediction. The results demonstrate the potential of parameter estimation over state estimation, revealing enhanced ENSO prediction skills achieved through more accurate parameter estimates. This work is interesting and worthy of publication. However, some minor revisions are necessary before acceptance.*

**Reply:**

We sincerely thank the reviewer for the suggestions and comments that help us improve the quality of our manuscripts.

*Major Comments:*

*Starting from line 141, the authors conducted sensitivity experiments by perturbing multiple parameters to assess the model's temperature and salinity variables' sensitivity to those parameters. It is noted that parameters were perturbed simultaneously. Have the authors considered perturbing these parameters individually? Could the sensitivity of variables to different parameters differ?*

**Reply:**
That's a good question. In fact, we have done previous experiments where we perturbed a single parameter and calculated the average ensemble spread of the results of the sensitivity experiments with the following results:

In these experiments, we still used the same initial field, and perturbed parameters to integrate the model. In these experiments, we perturbed only one parameter at a time, while keeping the other parameters constant. Thus, calculating the ensemble spread of the integration results allows us to assess the sensitivity of the model variables to this perturbed parameter.

[Figure]

Figure 1. Evolution of ensemble dispersion obtained after slightly perturbing different parameters and integrating the model for a period of time.

Figure 1 shows the global average of the ensemble spread of the SST with integration time, and the different coloured lines represent the integration results of perturbing different parameters. It is not difficult to find that the dispersion of the integration results of perturbing the four parameters is not much different, which indicates that the sensitivity of the temperature variables to the four BVDCs is not much different. The situation is similar for salinity. So we chose the scheme of simultaneous perturbation in the sensitivity experiment.

We put this conclusion in the discussion section of the sensitivity experiment results.

Thanks for the suggestion and comments.

*From line 188, the authors mention that during the first phase of Parameter Estimation (PE), parameters were perturbed but only State Estimation (SE) was used, lasting for a year. The state variables employed for PE and the motivation for this approach need clarification.*

**Reply:**

The motivation of this strategy is indicated by section 2.4 (experimental design and verification data). Zhang et al. (2012) showed that the signal-to-noise ratio of the state-parameter error covariance in a coupled model can be significantly improved after the state estimation reaches quasi-equilibrium. Thus, using the observation-constrained states that have reached equilibrium can effectively improve the accuracy of CPE (Zhang 2011a, b). In the PE experiment, we utilized perturbed parameters to perform state estimation in the beginning. After about 1 year, the model's errors in the state variables, especially temperature and salinity, were significantly reduced, and we assumed that the state estimation process had roughly reached equilibrium, and then enabled the parameter changes.

We have indicated this in the lines in section 3.2 as you pointed out.

Thanks.

*In Figure 8, the authors present Root Mean Square Error (RMSE) without specifying the reanalysis data it pertains to. Despite earlier indications of similar results from different reanalysis datasets, it is advisable to explicitly mention the data used.*

**Reply:**
We have calculated the RMSE for the analysed field and EN4 data in figure 8 and figure 9, which we have pointed out in the revised manuscript.

Thanks.

*Additionally, line 219 asserts that the maximum error occurs at the depth most sensitive to parameters, which is not immediately apparent. It is recommended to include a subfigure depicting parameter sensitivity along the equatorial range, using a logarithmic depth coordinate. Similar concerns are noted in Figure 9.*

**Reply:**
It is a very good suggestion, and we have modified figure 8 accordingly. Inside the new figure, we find a high degree of matching between the error of the SE results and the ensemble spred of the sensitivity experiment results, which highlights our conclusions even more.

As for Figure 9, we have revised the whole figure based on another reviewer's comments. Since the improvement in Atlantic is not due to parameter sensitivity but based on other mechanisms, we do not include the sensitivity results in this figure. Thank you for your suggestion.

*Minor Suggestions:*

*In lines 56-58, apart from the atmosphere, ocean, land, and sea ice, CESM encompasses other modules as well. The authors should use "as well as other modules" to accurately depict the model.*

**Reply:**
Changes have been made as you suggested, thanks.

*Is Equation (1) valid outside the Banda Sea region, using a value of 1.0 within the Banda Sea? Clarify this description for improved understanding.*

**Reply:**
Yes, the default background vertical diffusivity parameter is 1.0 in the Banda sea. We have re-written this part and clarified the description, thanks.

*Regarding line 108, the authors mentioned "daily profiles were merged and assigned to the final day of each sequence". Is there any other references employed the same approach to process the data?*
**Reply:**

In our previous study, e.g., Chen et al. (2022,2023), In our previous study, we used the above methodology to process the profile data. The main reason for this is that the EN4 profile data are unevenly distributed in both space and time, and if they are not processed using the appropriate method, overfitting may have occurred due to the

assimilation of too much data. Previous studies have shown that this processing method can produce effective state estimation results.

We have cited these papers in the manuscript. Thanks.

- Chen, Y., Shen, Z., & Tang, Y. (2022). On Oceanic Initial State Errors in the Ensemble Data Assimilation for a Coupled General Circulation Model. Journal of Advances in Modeling Earth Systems, 14(12). https://doi.org/10.1029/2022MS003106
- Chen, Y., Shen, Z., Tang, Y., & Song, X. (2023). Ocean data assimilation for the initialization of seasonal prediction with the Community Earth System Model. Ocean Modelling, 183(102194).

*Line 109 mentions interpolation to 31 layers. Could the specific depths of these layers be provided?*
**Reply:**

Specific vertical depths were obtained from the EN4 analysis data. We've added that to the draft, thank you.

Finally, we would like to thank the reviewers again for their valuable suggestions, which helped us to refine the details of our experiments and to better present the relevant results. Thanks.

---

## Author Comment (AC2)

** reviewer's comments in black and italics Author's comments in red

Review 2:

*This study investigates parameter estimation (PE) in improving climate forecasts of a coupled general circulation model by adjusting the background vertical diffusivity coefficients in its ocean component. Comparing the model states between the PE experiment and a state estimation (SE) experiment reveals that PE can significantly reduce the uncertainty of these parameters and improve the quality of analysis. The forecasts obtained from PE and SE experiments further validate that PE has the potential to improve the forecast skill of ENSO. The work is interesting and may has potential application in practice. However, there are some issues unclear in the manuscript. I would suggest that it should be published in* **GMD,** *subject to* **major revisions.**

**Reply:**

We sincerely thank the reviewer for the suggestions and comments that help us improve the quality of our manuscripts.

*1. Major concerns:*

*(1) The manuscript focuses on improving the background vertical diffusivity and its impact on ENSO forecast. However, the background vertical diffusivity is only important in the deep ocean and the coefficients of the KPP dominate the mixing process of the upper ocean. Therefore, results show that the main differences between PE and SE are in the deep ocean, and the improvement on the upper ocean is not significant as the authors mentioned. From the text, the author pays more attention to the process of ENSO which is mainly associated with the upper ocean process. So why do the authors not adjust and optimize the parameters of KPP?*

**Reply:**

Thanks to the reviewer for the comment, it is a very good one. We have revisited the motivation of this paper and emphasised the importance of parameter estimation of background vertical diffusivity coefficients within the ocean.

*In many OGCMs, vertical mixing can be parameterized separately by region, including upper boundary layer schemes and a diapycnal mixing scheme for the ocean interior. The K-profile parameterization (Large, 1994) is widely used to parameterize vertical mixing in ocean models. It includes a background diffusivity parameter that determines the diapycnal mixing in the thermocline. It is critical to the heat transfer between the upper boundary layer and the ocean interior. The*

*background diffusivity is typically set to a constant value, and its magnitude is determined by fitting the model to observations or theoretical considerations. As identified by much of the previous work, the background diffusivity parameterization is a key factor in vertical mixing parameterizations that have significant uncertainties and contribute to a large bias in SST simulations (e.g., Jochum et al., 2013; Melet et al., 2013; Zhang & Zebiak, 2002, 2004; Zhang & Gao, 2016; Zhu & Zhang, 2017). Zhu et al. (2018) have shown a better background diffusivity parameterization leads to more realistic simulations of the cold tongue and equatorial thermocline, which has the potential to affect the fidelity of simulated seasonal to interannual variability in the tropical Pacific, such as the ENSO phenomenon.*

We have put that previous paragraph in the revised paper to illustrate the motivation for this paper. However, according to Large (2003) and related studies, background diffusivity coefficients are parameters of the K-profile parameterization, which is also reflected in the namelist of CESM/POP. In this paper, we have focused on the parameter estimation of the background vertical diffusivity coefficients without including the upper boundary layer parameterization. The latter of course has some uncertainties and may be able to improve the simulation, which we will discuss in a future study.

Thanks for the comments.

*(2) The key issue of adjusting model parameters with real data is how to ensure the physical meaning of the parameters. The manuscript does not explain how to avoid the parameters from exceeding their physical values, specially for the spatial varying parameters.*

**Reply:**
Thank you for your question, which is very important for implementing parameter estimation experiments.

We apologize for the lack of precision in describing the parameters in the draft, which may have led to some misunderstandings. We have made major changes to this section, see section 2.1. Specifically, the parameters we estimate are just four constants that constitute the background diffusion parameters for latitudinal variations through Eq. (1). This latitudinally varying background diffusivity parameter can also be seen in Figure 6a.

[Figure]

Figure 6 (a) Default latitudinal structure of background vertical diffusivity.

In our experiments, we use the observations to update these four constants. We first perturb the parameters to conform to a Gaussian distribution using a standard deviation of 30% of their values, which adds uncertainty to the parameters and is practically unlikely to result in unphysical values. The parameter estimation process is then used to reduce the uncertainty in the parameters, and it will adjust the ensemble of parameters based on real observations and essentially not run out of reasonable ranges: i.e., the values of the four parameters all match their magnitudes. We examined the data assimilation results, which turned out to be the case.

Of course, for the sake of the integrity of the algorithm, we need to use certain control conditions to avoid unphysical parameter values for a certain ensemble member (although very rare). So, we added the following statements in the description of the method.

*It is worth noting that in order to avoid unphysical parameter values, after each parameter estimation, if an abnormal parameter value (e.g., negative value) occurs for an ensemble member, we remove the parameter and use the parameter of a neighbouring member to integrate the model.*

(3) It is unclear how the adaptive spatial averaging is calculated. How big is the specific average bin? Does it cause spatially discontinuous?

Reply:

We apologise for missing some details. We have added the details of the method in the revised version, see section 3.2.

*In each data assimilation step, we transformed each parameter from a single scalar value into a two-dimensional field, considering spatial dependence and localization during the assimilation. Afterward, we use the adaptive algorithm to average the two-dimensional parameter fields, to produce a scalar value incorporated in subsequent model integration. This algorithm calculates the ratio of the a posteriori standard deviation to the a priori standard deviation at each grid point after each update of the two-dimensional parameters, which implies the strength of the effect of assimilation, and then averages the parameter values at grid points where the ratio exceeds a certain threshold. This threshold is chosen using an adaptive algorithm to ensure that a certain number of grid points (in this experiment 10,000 out of a total of 80,000 grids) are included in the calculation of the averaged parameters. More details refers to Shen and Tang (2022).*

No spatial discontinuity occurs because the four constant-valued parameters are still obtained after applying spatial averaging to the two-dimensional parameters.

*(4) Table 1 involves 4 parameters and the use of parameters is regional. Do you only adjust the corresponding parameters in specific regions such as the equator and the Banda Sea?*

Reply:

*We have answered this questions in the response to Major concerns (2). Once again, we apologise for the misunderstanding caused by inappropriate descriptions.*

*(5) How are the 20 sensitivity experiments and other experiments set up and operated? A figure regarding this should be added. Based on Table 2, the parameter adjustment is quite small, and the parameters tend to reach a certain equilibrium state. However, the observations (number and locations) used each year are different. Why do the adjustment of parameters not change year by year?*

Reply:

Thanks to your suggestion, we have drawn a schematic to illustrate the experimental framework for the sensitivity experiment, the state estimation experiment and the parameter estimation experiment.

*The schematics in Figures 2a-c show the sensitivity experiment, the SE experiment, and the PE experiment, respectively. It can be seen that the sensitivity experiment is a free integration experiment using the same initial condition and different parameters. The SE experiment uses the ensemble of state variables and the same default parameters. At the same time, the PE experiment uses ensembles for both*

*state variables and parameters. PE experiments are divided into three phases, which we will specify in the Results and Discussions section. Moreover, it also shows that the state and parameter estimation results are used in the later hindcast experiments.*

[Figure]

Figure 2. The schematic diagrams of the sensitivity experiment (a), SE experiment (b) and PE experiment.

For the second problem, since we use the spatial averaging scheme described above, the parameter values changed per assimilation are very small. And data assimilation is a process that reduces ensemble spread, which is inevitably reduced even if we use the covariance inflation method described earlier. Moreover, since the ensemble spread of the state variables grows with model integration, and the parameters are generally assumed to be constant during integration, the parameter ensemble must eventually degenerate - the spread becomes 0, and all observations are rejected. In practical assimilation, the conditional covariance inflation (CCI) method can ensure that the ensemble spread does not fall below a lower bound (Liu et al. 2014). The CCI is designed to inflate the parameter ensemble spread back to a predefined threshold value when it is smaller than the threshold. Details on this method can also be found in our previous work (Shen and Tang, 2022), which is an ideal experiment using the same CESM model.

However, in this work, we intentionally do not use the CCI method and let the parameter ensemble degenerate. At this point, all ensemble members can use the same improved parameters to carry out the ensemble forecast, which makes the ensemble forecast easier to carry out and compare with other schemes. This may not be the optimal solution for parameter estimation, but it is the most convenient solution for carrying out realistic forecasts.

The experiment in Figure 3 also assimilates synthetic TS profiles and SST observations to estimate the vdc1 parameter in the same CESM model. As shown in the figure below, the amount of change in the parameter values after a period of assimilation is very small as long as the proper parameter covariance expansion coefficients are used (scenarios a - c), so we believe that switching off the CCI also yields relatively improved parameters.

Thanks for the questions and suggestions.

[Figure]

Figure. The evolution of the parameter mean (red solid line) and spread (red dotted lines) over the data assimilation period using extra parameter inflation with (a), 1.2 (b), 1.3 (c), and 1.5 (d), respectively.(Shen and Tang, 2022)

*(6) Line 215: "The highest error is observed in the sea surface layer, ...." I can't see the relevant results in Fig.7. The main differences are still at the depths of 600m-1200m in the Atlantic and tropical regions.*

Reply:

Sorry for the lack of clarity in the sentence.

For the original Figure 7 (Figure 8 in the revision), we refer to the fact that the largest salinity errors in both SE and PE analysis fields occur at the sea surface, as can be seen from the larger RMSE in the surface layer in Figure 7. It is certainly true, as you say, that the most significant improvements in PE over SE are in the deep tropics as well as in the deep Atlantic. The reasons are explained in the next two figures.

[Figure]

Figure 7. The salinity RMSE of the data assimilation results with EN4 (top), ORAS4 (middle), and ECDA (bottom) for the period of 2008-2017 by region. The regions are - Global (within 66.5°N-S, Pacific, Indian Ocean, Atlantic, and intra-tropical (within 30°N-S), respectively, from left to right.

*(7) Based on the sensitivity experiments, the main impact is in the upper 100m. However, from the assimilation results, the main differences are still at the depths over 600m in the tropical and Atlantic regions. Why is this?*

Reply:

Referring to another reviewer's comment, we redrew the original Fig. 8 to show the correspondence between parameter sensitivity and SE analysis error.

[Figure]

Figure 9. The temperature (a) and salinity (b) RMSE of the SE results and EN4 data in the tropics; the mean temperature (c) and salinity (d) spreads of the sensitivity experiment results in the tropics; the difference between the temperature (e) and salinity (f) RMSE of the SE results and that of the PE results, respectively.

*As Figure 9a shown, the most considerable temperature errors appear in all oceans around the depth of 100 meters, which matches the sensitivity analysis result for temperature depicted in Figure 9c. Figure 9e denotes an improvement in PE for these errors, implying its usefulness throughout the tropics. It is noted there is improvement in areas where the temperature are sensitive to the uncertainty of those parameters. And the temperature error in the deeper layers of the tropical oceans has also been reduced, especially in the deeper Atlantic Ocean. Figures 9b, 9d and 9f show the results for salinity. Although not as significant as temperature, the depths and areas where salinity errors emerge in the SE analysis align with those sensitive to parameters.*

The story for the Atlantic is quite different. Danabasoglu et al. (2012) show the zonal-mean potential temperature $\theta$ and salinity $S$ of CCSM4 (which uses the same ocean model as CESM) minus climatology from observavation in the following figure. They noted that the deep Atlantic Ocean remains generally warmer than observed by about 0.58C in the mean. The local $\theta$ and $S$ maxima between 20º and 30ºN at a depth of about 1000 m are associated with the warmer and saltier than observed Mediterranean outflow through the Strait of Gibraltar. The largest salty biases occur in the deep Atlantic Ocean. The upper-ocean Atlantic north of 15º N remains mostly saltier than the climatology.

[Figure]

Figure (Danabasoglu, 2012) Zonal-mean (left) potential temperature (ºC) and (right) salinity (psu) CCSM4 minus PHC2 observations difference distributions. (top to bottom) The global, Pacific, Indian, and Atlantic Ocean differences are shown.

We draw a similar climatology bias for temperature and salinity in Figure 10. Although compared to different observations, similar results are demonstrated,

especially as the deep ocean also has a warm and salty bias caused by systematic biases in the model. By comparing the climatology bias of the results from two experiments, it is seen that the most significant improvement of the PE on the SE lies in the Atlantic Ocean at a depth of 1000 m between 20° and 30° N latitude. It strongly suggests the contribution of improved background diffusivity parameters to reducing model systematic biases.

It can be inferred from the conclusions of Danabasoglu et al. (2012) that this improvement also stems from the improvement of the outward flow in the Strait of Gibraltar. Although the sensitivity and analytical errors in this region cannot be demonstrated directly due to resolution, the effect is demonstrated by affecting the 1000m Atlantic Ocean between 20° and 30° N latitude.

We have revised the disscussion in section 3.3, thanks for the comments.

[Figure]

Figure 10. The zonal-mean temperature (a) and salinity (b) minus the climatology calculated from EN4 data using t SE results, and The zonal-mean temperature (c) and salinity (d) bias using the PE results.

*(8) Line 221: "It is noted there is substantial improvement in areas where the errors are significant, except for the deep Atlantic Ocean". Estimated from the magnitude of SE-PE, the proportion of error reduction is less than 5%. It can't be said "substantial improvement".*

Reply:

We agree with the reviewer's comments, and since we have revised this part of the discussion, this statement has been amended, thanks.

*(9) Fig.9 only mentions the differences in the Atlantic Ocean. How about the tropical ocean?*

Reply:

The original Figure 8 (Figure 9 in the revision) displays the RMSEs of the SE experiment and EN4 data in the tropics while emphasizing the disparity with the PE experiment. We have shown the temperature and salinity RMSE of the SE results and EN4 data in the tropics; the mean temperature and salinity spreads of the sensitivity experiment results in the tropics; the difference between the temperatureand salinity    RMSE of the SE results and that of the PE results, respectively.

Please see the response to Major concerns (7).

*(10) The forecast of ENSO does improve to a certain extent, but the improvement is not significant (The change of RMSE is less than 0.1°). Shown From Fig.11, it seems that most differences locate in the equatorial region. It may connect to the reduction of diffusivity in the region (smaller mixing and higher SST?). The authors should give more explanations regarding this issue.*

**Reply:**

This is a very good commet and is important for the application of the results in this paper.

In response to the major concern (1),    we have re-emphasised the significance of parameter estimation for BVDCs in the revised version. Specifically, Zhu and Zhang (2018) have demonstrated that an enhanced background diffusivity parameterization results in more realistic simulations of the cold tongue and equatorial thermocline. This improvement holds the potential to impact the accuracy of simulated seasonal to interannual variability in the tropical Pacific, including phenomena like ENSO.

The outcomes of our data assimilation experiments reveal a notable reduction in analysis error within the thermocline of the equatorial Pacific due to parameter estimation, as illustrated in Figure 9. This, in turn, yields improved initial conditions for ensemble prediction. Simultaneously, the refined parameters contribute to enhanced global vertical mixing. Consequently, the SSTA in the tropical Pacific, derived from our forecasts incorporating parameter estimation, exhibit a higher ACC with observational data, as depicted in Figure 12.

In addition, since the advantage of the PE analysis field in the subsurface layer takes some time to affect the SST, the forecast skill based on the PE results is only significantly greater than that of the SE results at longer forecast times.

Of course, as the reviewer points out, the correlation coefficients between the nino indices computed with the PE forecast results and the indices computed with

observations are only slightly larger than in the SE case. It can be concluded that PE only slightly improves the prediction skill of this coupled prediction system. However, the prediction skill of ENSO is affected by many factors, such as predictability, in addition to the initial conditions and model errors (Liu et al. 2021). Therefore every bit of improvement of the ENSO dynamical prediction skill is of some practical significance. Nevertheless, we will also pursue higher dynamical prediction skill in the future research.

We have added the assertion about the advantages of PE in prediction and the reasons for it in the Results and Discussion section, and added the last part of the discussion in the Conclusions section.

Thanks for the suggestion.

*2. Minor Concerns:*

*(1) The assimilation just involves the ocean component. It seems not a couple assimilation system.*

**Reply:**

I'm sorry we didn't go into all the details of the data assimilation system. The CESM model of our application enabled all the components, i.e., it is a fully coupled earth system model.
A recent review by Zhang et al.(2020) gives the definition of coupled data assimilation (CDA). That is, the DA process is performed within the coupled model directly in CDA. So, although we assimilate only ocean observations in order to focus on the ocean parameterisation scheme, this also falls within the framework of coupled assimilation as we integrate the fully coupled model. This is often referred to as weakly coupled data assimilation (WCDA).

We have pointed out this in section 2.2. Thanks.

Zhang, S., Liu, Z., Zhang, X., Wu, X., & Deng, X. (2020). Coupled data assimilation and parameter estimation in coupled ocean–atmosphere models: A review. Climate Dynamics, 54(11).

*(2) In Fig.7, is it the ORAS5 or ORAS4?*

**Reply:**

It is ORAS4, we have made changes in the revision. Thanks for pointing that out.

*(3)Line 116: "These factors were determined empirically and verified in prior studies." References should be added here.*

**Reply:**

We added our previous paper as a reference.

*(4) Units should be added in Figs.6, 7 and 10.*

**Reply:**

Changes have been made as suggested. Thank you.

Finally, we are very grateful to the reviewers for their comments and suggestions, which were very meaningful in refining the details of the paper and explaining its results.

---

## Author Response (AR2)

Dear Editor,

We have added suggested paragraphs to the discussion section as requested by the reviewers (see lines 327-334). Once again, we thank the reviewers for their insightful comments and suggestions, which greatly improved our manuscript. Many thanks also to the editors.

Zheqi Shen and coauthors,